# Simulation-Based Education in Trauma Management: A Scoping Review

**DOI:** 10.3390/ijerph192013546

**Published:** 2022-10-19

**Authors:** Blanca Larraga-García, Manuel Quintana-Díaz, Álvaro Gutiérrez

**Affiliations:** 1Escuela Técnica Superior de Ingenieros de Telecomunicación, Universidad Politécnica de Madrid, 28040 Madrid, Spain; 2IdiPAZ, Hospital La Paz Institute for Health Research, 28029 Madrid, Spain

**Keywords:** education, clinical simulation, trauma management

## Abstract

Trauma injuries are an important healthcare problem and one of the main leading causes of death worldwide. The purpose of this review was to analyze current practices in teaching trauma management using simulations, with the aim of summarizing them, identifying gaps and providing a critical overview on what has already been achieved. A search on the Web of Science website for simulation-based trauma training articles published from 2010 onwards was performed, obtaining 1617 publications. These publications were screened to 35 articles, which were deeply analyzed, gathering the following information: the authors, the publication type, the year of the publication, the total number of citations, the population of the training, the simulation method used, the skills trained, the evaluation type used for the simulation method presented in the paper, if skills improved after the training and the context in which the simulation took place. Of the 35 articles included in this review, only a few of them had students as the target audience. The more used simulation method was a high-fidelity mannequin, in which the participants trained in more technical than non-technical skills. Almost none of the studies introduced an automated evaluation process and most of the evaluation methods consisted of checklists or questionnaires. Finally, trauma training focused more on treating trauma patients in a hospital environment than in a pre-hospital one. Overall, improvements in the evaluation method, as well as in the development of trauma training on undergraduate education, are important areas for further development.

## 1. Introduction

Trauma injuries are responsible for 9% of global mortality and are considered a risk all around the globe. These injuries result from traffic collisions, drowning, poisoning, falls or burns and violence, causing more than five million deaths worldwide annually [1]. Moreover, a large number of those injuries cause temporary or permanent disabilities, incurring important consequences on the patients’ lives. Therefore, a fast identification and management of trauma injuries is of great importance. To do that, a systematic and rapid approach should be applied [2].

The Advanced Trauma Life Support (ATLS) course was created in 1978 by the American College of Surgeons, and it is currently taught in over 60 countries [3]. This course has used a variety of simulation modalities to teach trauma management and, since then, other trauma management courses have arisen [4,5,6,7]. Additionally, regarding pre-hospital trauma management, the two courses mainly referenced being the Pre-Hospital Trauma Life Support (PHTLS) course [8,9,10] and the International Trauma Life Support (ITLS) course [11,12], which focus on education for first responders and also use simulations, as well as classroom sessions. Trauma training focuses on several aspects that can be classified into technical skills and non-technical skills. Technical skills refer to the application of a correct triage, primary and secondary surveys, including the techniques and treatments needed to achieve that. Non-technical skills focus on communication, leadership, management of situations and decision making. Even though both types of skills are intrinsically related, some training focuses only on technical skills, others in non-technical skills and others on both.

In this context, it is important to highlight the role of clinical simulations. Clinical simulations started to support clinical training by taking into account patient safety [13,14,15,16], but it also offers some other benefits, such as the opportunity to repeat a simulation as many times as needed, or to train a great variety of technical and non-technical skills [17,18,19,20]. Nevertheless, there is still limited evidence on the impact of simulation-based training on the performance in trauma management [14,16] and on the long-term knowledge retention of such training [16,21]. Clinical simulators are classified according to the concept of fidelity, with the simulated model’s relation to its closeness to reality being the main classification: low-, medium- and high-fidelity simulators. Low-fidelity simulators are anatomical representations of a part of the body to train simple tasks and to acquire basic motor skills to be able to develop those tasks. It is generally composed of low technology. Medium-fidelity simulators integrate low-complexity software programs that allow manipulating physiological variables to assess knowledge during decision making in environments such as cardiopulmonary resuscitation. Finally, high-fidelity simulators are life-size mannequins that integrate mechanical devices and computer technology to train advanced techniques and skills in handling critical situations. In principle, high-fidelity simulations are the best option; however, according to [22,23,24], there is no important difference with respect to knowledge and skill improvements of high-fidelity compared to low-fidelity simulators. In [22], the skill performance evolution comparing low-fidelity and high-fidelity simulations is studied in a systematic review. This study shows that, in the short-term, the use of high-fidelity simulators provides a moderate benefit compared to low-fidelity; however, in the long term, no benefits are obtained. Additionally, in [23], a study in simulated neonatal resuscitations is presented. It shows no differences after training with a low-fidelity or a high-fidelity simulator. Finally, a randomized control trial with more than 100 undergraduate students was conducted using low- and a high-fidelity simulators [24]. The conclusion was that the improvement obtained was similar after both training courses.

### Goals of This Investigation

The purpose of this review was to analyze the current practice in teaching trauma management using simulations with the aim of summarizing them, identifying gaps and providing a critical overview on what has already been achieved in terms of trauma training.

The secondary goals were to provide specific gaps with respect to the target audience, simulation methods used in trauma training, types of skills trained and evaluation methods used to measure knowledge acquisition.

## 2. Materials and Methods

### 2.1. Data Sources

A search was performed on the Web of Science website. This website provided access to the following databases: the Web of Science Core Collection, BIOSIS Citation Index, BIOSIS Previews, Current Contents Connect, Derwent Innovations Index, KCI-Korean Journal Database, MEDLINE, Russian Science Citation Index and SciELO Citation Index. The search was performed using the topic searching field. This topic field included the title, the abstract and/or the keywords, and the terms used in the search were as follows: simulation OR web simulation OR patient simulation OR mannequin OR interactive AND trauma AND training OR education.

### 2.2. Study Selection

This initial search provided 1617 publications, of which 7 were duplicates, as shown in Figure 1. Then, titles of the 1610 articles were screened, removing those which were not within the scope of this review. Therefore, the ones that focused on children, adolescents, post-traumatic stress disorder (PTSD), obstetrics and other specialties, which were not traumatic injuries, were excluded. Moreover, articles published from 2010 to 2021 were selected, obtaining 120 articles. Subsequently, the 120 articles were reviewed, including their titles and abstracts, finding that 55 articles were, in fact, out of scope. These 55 articles were out of scope according to the same logic already used: excluding focus on children, adolescents, PTSD, obstetrics and other specialties that were not trauma-related. Within the titles of these articles, this was not detected; however, when going through the abstracts, this was perceived. Hence, 65 articles were reviewed and analyzed. From these 65 articles, 17 were review articles and 13 were still out of the scope, as they either focused on a very specific technique or they considered simulations in a different field, with no focus on trauma. Therefore, 35 articles really focused on trauma training and provided studies on how different simulation training techniques could impact trauma management training. The process followed is shown in Figure 1, following the PRISMA flow diagram and according to the recently published PRISMA Guidelines for Scoping Reviews (PRISMA-ScR) [25].

### 2.3. Data Analysis

The 35 studies included in this review were analyzed in a specific template that was developed for that purpose. In that template, the following information was gathered: authors, publication type, year of publication, total number of citations, population of training, simulation method used, skills trained, evaluation type used for the simulation method presented in the paper, if skills improved after the training and the context in which the simulation took place. One author (B.L.-G.) drafted this structure and an initial data analysis. Then, it was discussed with two other authors (A.G. and M.Q.-D.), and the final template with the final structure was produced and completed after a thorough study of all the articles.

## 3. Results

### 3.1. Characteristics of the Study

The main characteristics of the 35 articles included in this scoping review are shown in Table 1.

### 3.2. Main Results

#### 3.2.1. Target Audience of the Training Courses

Only 7 out of the 35 studies focused on medical students, as shown in Figure 2a. From these seven studies, one of them focused on paramedic students and another training course focused on both medical students and doctors together. This showed that only 20% of the studies presented a simulation-based trauma training course delivered specifically for medical students during their undergraduate academic training. The rest of the studies presented simulation-based training for both consultants and residents in a similar proportions, and just three of them had paramedics as the target audience. The size of the target audience trained varied from 18 [31] to 444 people [35].

#### 3.2.2. Simulation Methods Used

With respect to the simulation methods used during the simulation-based training courses analyzed, 18 of them (51.4% of the training courses) used high-fidelity mannequins during the trainings, as shown in Figure 2b, whereas the rest used any other methods. Regarding the other methods used, one of them presented simulation cards as the training method used; three of them used standardized patients trained for that purpose; another three studies used skill stations to practice several skills during the trauma management training; and seven of them used virtual reality training as the simulation method. This virtual reality modality included an immersive experience, in which VR goggles and a virtual reality scenario were involved. Only two cases considered a desktop virtual setting, in which a virtual patient was assessed. The remaining studies did not specify the simulation method used.

#### 3.2.3. Types of Skills Acquired after the Training Courses

Regarding the skills that the trainees gathered after the trainings, 18 of them (51.4% of the training courses) focused on training technical skills considering the application of correct protocols to attend to trauma patients, as well as specific treatments and techniques for trauma treatments. Only two of the studies included in this scoping review focused on non-technical skills, and eight of them focused on both technical and non-technical skill training, as shown in Figure 2c.

#### 3.2.4. Evaluation Methods Used

Taking into consideration the evaluation methods, only two of the studies provided an automated evaluation of the training delivered, as shown in Figure 2d. It was considered an automated evaluation of the training when the simulation method used provided an evaluation automatically taking into account the performance of the training courses. To achieve this, simulation methods should be prepared to gather all necessary information for such an evaluation. The rest of the studies included in this review provided an evaluation of the trauma training that was not automatically obtained. With respect to the evaluation methods used in the different trauma management training courses, a more comprehensive analysis was performed. From the different methods presented in the 35 articles, 18 of them used either checklists or evaluation forms that were previously prepared, providing different options to the trainees. Then, there were four studies that used subjective evaluation methods, which included interviews, written comments or direct observations. Ten of the studies used both checklists and subjective evaluation methods and, finally, three studies did not state the evaluation method used, as shown in Figure 2e. The trainers were usually experienced surgeons, emergency physicians, critical care specialists or specialized instructors from training courses such as the ATLS or the Advanced Cardiovascular Life Support (ACLS) course. Additionally, some training took place in simulation centers; therefore, the evaluations were compiled from the members of these centers.

#### 3.2.5. Context of the Simulations

With respect to the context in which the simulations took place, four of them focused on extra-hospital training, presenting trauma management training courses that focused on extra-hospital scenarios, in which the personnel and the resources are different from the ones in the hospital. Additionally, 25 studies focused on hospital trauma management, whereas five of them, as shown in Figure 2f, provided trauma management training with focus in both extra- and in-hospital scenarios.

#### 3.2.6. Limitations

As stated in this section, the number of articles was limited by the searching and eligibility criteria. This scoping review was limited to simulation-based trauma training, in which the terms used were certain simulation methods that may not have gathered a complete representation of trauma training, but still the majority. Moreover, the search was limited to articles in English and published from 2010 onwards. This was conducted as technology has improved in the past decade and new simulation methods have arisen. Therefore, it was considered that the results obtained during this period would better reflect the current situation of simulations in trauma training. Another limitation was that, in most cases, there was no comprehensive explanation of all the details within the simulation-based trauma training and, in some cases, the pilot studies involved a reduced number of trainees.

## 4. Discussion

Taking a look at the results obtained for the types of populations that received the trauma training, it stood out that there was scarcity in trauma management training courses for medical students. Moreover, according to [16,57], the best simulation method and procedure to teach trauma management to medical students have not yet been established. This is a field that needs further development, as medical students should be trained on trauma management skills considering that they are soon-to-be residents. As residents, they are going to be the first attendants in the hospital; therefore, having specific trauma training would allow for better treatments for patients [20,30,58]. Additionally, trauma training supports clinical reasoning learning. This is key for clinical practice, and could be obtained with trauma management training [48].

According to Lewis C. and Veal B. [21], none of the 29 articles included in the review could demonstrate a significant objective impact on the mortality and morbidity of trauma patients; therefore, there is still more research needed in this field. Nonetheless, there are studies [29,30,59] that support and present statistical improvements in trauma management performance after simulation training. They confirm that, if the correct simulation modality is used, the expected outcome of the patient could be more easily predicted [2]. Therefore, it is important to know the different simulation modalities and how they should be implemented within trauma management training. In [2], it was stated that trauma training uses both low- and high-fidelity training modalities. Low-fidelity training allows to reproduce and practice technical skills such as airway management, whereas high-fidelity training offers the possibility to train both technical and non-technical skills. Additionally, standardized patients could also be used to train non-technical skills and, if properly garbed with the appropriate modules, some technical skills could also be practiced. Moreover, virtual reality is currently increasing its presence, as it allows to connect multiple users at multiple locations, increasing availability to centers with limited resources. This simulation modality offers the possibility to immerse learners within authentic clinical scenarios at a low cost.

With respect to the simulation methods used, traditional simulation methods try to imitate real patient simulations. That is the reason why high-fidelity mannequins or actors have been widely used [27,60]. Nevertheless, technology allows for the development of other simulation methods that could offer solutions to the limitations that actors and mannequins have. Simulations with actors have limitations, as some techniques cannot be applied. High-fidelity mannequins are expensive models that require specific technical requirements and resources. Virtual reality offers a solution to these limitations, as it allows trainees to immerse themselves in the situation, enabling them to accomplish different trauma scenarios without compromising the patient and allowing institutions to train a large number of trainees [26,47,48,61]. However, each simulation method has its advantages and disadvantages and, therefore, a further reflection is needed with respect to the selection of simulation method, as stated in [2,16]. It is also important to consider, for the selection of the simulation method, which skills to train.

Taking into account the results obtained for the skills trained, there was still a majority of simulation-based training courses that focused on technical skills. The goal of these training courses is to teach complex and specific skills [53]. Moreover, the number of training courses that consider non-technical skills is increasing [43,62,63]. That is the reason why the number of simulation-based training courses that consider both types of skills is also higher. It is important to highlight that the articles included in this scoping review focused on individual training. Therefore, it makes sense that more articles focused on training technical skills. The training courses that focus on non-technical skills prefer training in groups or teams, as this allows to practice those skills. Consequently, and as previously highlighted, depending on the skills to train, one simulation method can be better than another.

According to [15], medical training courses that use simulations should be adapted to the level and the type of education. Therefore, this article considered undergraduate teaching, postgraduate teaching, continuing medical education, disaster management and military trauma management. Furthermore, according to [2], the training courses should focus on the types of skills to train. Therefore, this article proposed to classify training courses into either task-oriented or non-technical-skill-oriented training, independently on the individual level and type of education of the trainee. The main idea behind these studies is identifying the focus of the training and then trying to find a simulation method that fits better with that focus, independent of the name provided to the focus of the training. It is clear that the trend is to incorporate non-technical skills within training courses in order to create a comprehensive trauma program; therefore, this trend is generally perceived in trauma training. Consequently, high-fidelity mannequins seem to be the best option; however, incorporating virtual reality together with low-fidelity mannequins could be another alternative with some advantages, such as the cost of the chosen simulation method used. For trauma training, as technical and non-technical skills need to be trained, simulation methods that combine low-, medium- and high-fidelity training should be considered.

Regarding the evaluation methods currently used in simulation-based training courses, only two of them considered the option of having an automated evaluation method [31,51]; however, either this was only partially considered, or the details on how automation was achieved were not explained. Therefore, the majority of the training courses analyzed in this scoping review did not offer an automated evaluation method, showing an important gap. It is true that there is an important discussion about how the evaluation of simulation-based training must be conducted [18,20,27,28,38,39,44,48,51], but it is surprising that the majority of the articles did not even consider the option to include an automated option. Additionally, this was unforeseen, as the advantage of having high-fidelity mannequins or other simulation methods is that they allow for the possibility to gather objective information directly from them. That information would be extremely valuable, as it is entirely objective, which fits with the purpose of using the simulations to provide a more objective evaluation method [40,64]. Additionally, the objective information gathered with the simulation methods has a positive impact on trainees, as it provides high-quality feedback, which allows them to see the impact of their actions during the simulation. This supports skill learning and performance [32,65,66]. As the majority of the trainings did not use an automated evaluation method, an analysis on which methods were used was performed. Many of the training courses used written evaluation forms or checklists. They were specifically developed for the trauma training provided, as stated in [18,27,33,34,37]. This allowed for the evaluation process to be more objective, though not entirely, as the trainees’ answers to the questionnaires or checklists comprised their opinions on how the simulations occurred. That opinion is valid and necessary after a simulation-based training; however, evaluating the performance of the training only with this information should not be the case. Just four of the training courses used purely subjective evaluation methods that consisted of either personal interviews, written comments or evaluations conducted through direct observations. Finally, most of the articles analyzed in this scoping review focused on training either technical or non-technical skills for traumas that took place in a hospital environment; however, the presence of pre-hospital training is increasing [17]. This situation highlights the importance of training all professionals involved in trauma scenarios in any of the environments in which the patient could be located, considering that the resources and personnel available in each of the settings are different.

## 5. Conclusions

This scoping review showed that there is an important gap with respect to the current evaluation methods and the training of medical students on trauma management. There are currently discussions on how to better evaluate simulations; however, none of them focus on the benefits of including purely objective information that could be easily provided using simulations. Therefore, finding this gap creates opportunities for new lines of work to develop and to include this type of evaluation together with others currently in use. This could provide a more solid evaluation process. Additionally, including trauma training in medical students’ education could have important benefits as already highlighted, which should encourage medical schools in developing trauma training within their medical degrees.

With respect to the other gaps found, further work should be conducted on classifying simulation modalities depending on the focus of the trauma training. This would allow to investigate all the possible options considering, additionally, the technology evolution and the budget available. Additionally, pre-hospital settings should be included in trauma training courses.

## Figures and Tables

**Figure 1 ijerph-19-13546-f001:**
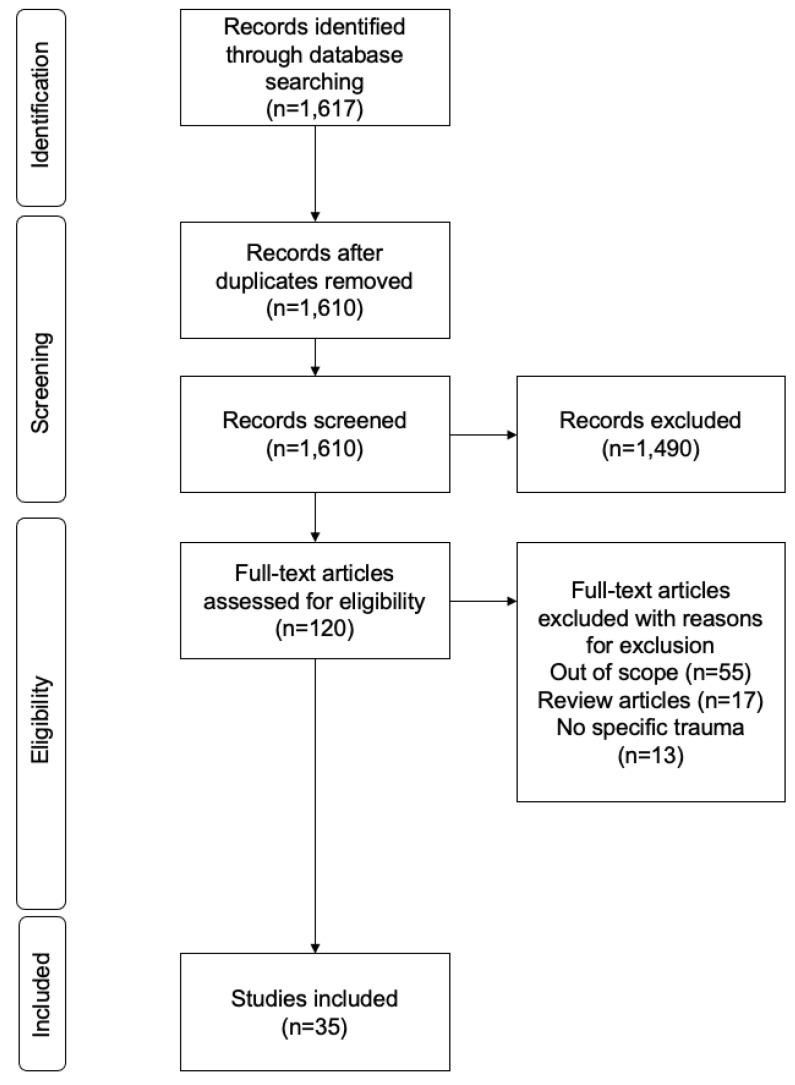
PRISMA flow diagram showing the study selection process.

**Figure 2 ijerph-19-13546-f002:**
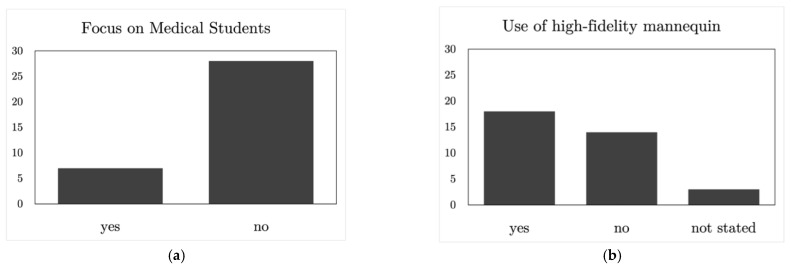
Results of the different aspects highlighted in this scoping review: (**a**) the focus on medical students of the trainings, (**b**) the use of high-fidelity mannequin simulators, (**c**) the type of skills trained, (**d**) if there is an automated evaluation process, (**e**) the type of evaluation used in the training, and (**f**) the context in which the simulation focuses.

**Table 1 ijerph-19-13546-t001:** Summary of the articles included in this scoping review.

Study	Population of the Training	Simulation Method Used	Skills Trained	Evaluation Type	Context	Publication Year	Number Citations
Patel et al. [26]	Residents	CineVR	NS	SB	IH	2020	1
Knudson et al. [27]	Residents	L and HF	T and NT	WE	IH	2010	146
Fernandez et al. [28]	Residents	LF and HF	T	WE	IH	2012	133
Cohen et al. [29]	Prehospital clinicians and emergency medicine consultants	VR	T	WE	PH and IH	2013	90
Ruesseler et al. [30]	Final year medical students	HF	T	SB and WE	PH and IH	2010	128
Harrington et al. [31]	ATLS trainees	VR	T	WE	IH	2018	76
Murray et al. [31]	Emergency medicine, surgery and anesthesia residents	HF	T	WE	IH	2015	36
Cohen et al. [32]	Ambulance HART practitioners, surgical residents and emergency consultants	VR	T and NT	SB and WE	PH and IH	2013	43
Amiel et al. [33]	Physicians and nurses	SS and HF	T and NT	WE	IH	2016	28
Pringle et al. [34]	Attending and senior resident physicians	SP	T and NT	WE	IH	2015	26
Jacobs et al. [35]	Surgeons	PS	T and NT	WE	IH	2010	25
Bredmose et al. [36]	Helicopter emergency medical service doctors and paramedics	HF	NS	SB	PH	2010	50
Springer et al. [37]	Residents	HF	NS	WE	IH	2013	18
Lennquist et al. [38]	Physicians, nurses, paramedics, military doctors and administrators	CC	T and NT	SB	PH and IH	2014	25
Jawaid et al. [39]	Final year medical students, interns and consultants	L, SS and CS	T	WE	IH	2013	19
Ali et al. [40]	Surgical residents	HF	T	SB and WE	IH	2010	14
Courteille et al. [41]	Medical students and residents	L and VR	T	WE	IH	2018	28
Aekka et al. [42]	Non-doctor first responders	HF	T	SB and WE	PH	2015	20
	Nurses, radiology technicians and attending and trainee physicians	HF	T	NS	IH	2018	26
Doumouras et al. [43]	Residents	HF	NT	WE	IH	2017	11
Figueroa et al. [44]	Interns	L, SS and HF	NS	WE	IH	2016	22
Sullivan et al. [45]	Residents and emergency nurses	HF	NT	WE	IH	2018	13
Kaban et al. [46]	Residents	NS	T	WE	IH	2016	9
Alsaad et al. [29]	Residents	HF and SP	T	WE	IH	2017	21
Taylor et al. [47]	Paramedics and different roles involved in emergency medicine	VR	NS	NS	PH and IH	2011	11
Fleiszer et al. [48]	Undergraduate medical students	VR	T	SB	NS	2018	15
Cuisinier et al. [20]	Medical students	HF	T	WE	IH	2015	4
Farahmand et al. [49]	Interns	L, CS and SS	T	SN and WE	IH	2016	11
Park et al. [50]	Residents	NS	T	NS	IH	2020	6
Walker et al. [51]	Residents	SP	T and NT	SB and WE	IH	2016	3
Hayden et al. [52]	Nurses, radiology technicians, attending and trainee physicians	HF	T	NS	IH	2018	49
Kuhlenschmidt et al. [53]	Residents	SS	T	WE	IH	2020	0
Cecilio-Fernandes et al. [30]	Medical students	HF and SS	T	SB and WE	IH	2019	3
Gräff et al. [54]	Doctors	HF	T and NT	SB and WE	IH	2017	4
Mills et al. [55]	Paramedic students	SP	NS	SB and WE	PH	2018	11
Campbell et al. [56]	Paramedics	HF	NS	SB and WE	PH	2018	1

NS: not stated; VR: virtual reality; SP: standardized patients; L: lectures; HF: high-fidelity mannequin; LW: low-fidelity mannequin; SS: skill stations; PS: porcine simulation; CC: casualty cards; CS: case scenarios; T: technical skills; NT: non-technical skills; WE: written evaluations or checklists; SB: subjective evaluation; PH: prehospital; IH: in-hospital.

## Data Availability

Not applicable.

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
