# Peer review of "Simulation-Based Education in Trauma Management: A Scoping Review"

_ijerph, 2022, doi:10.3390/ijerph192013546_

Round 1
Reviewer 1 Report
Thank you very much for giving me the opportunity to review your work. As you clearly point out, trauma injuries are an important health care problem, and it is important to train health care teams to adequately take care of trauma patients. In your review, you provide an overwiew of the current state of simulation-based education in trauma management, identifying possible gaps and suggesting improvement.
I have a few minor suggestions:
Introduction: provides a good overview over the topic and clearly states the goals of the study.
Materials and Methods: sound description of the process.
Results:
I would have liked a comment on the kind of VR (virtual reality) used in these studies. Was it fully immersive VR or some kind of desktop virtual setting?
Furthermore, I would have liked some information on where the study was conducted (high-income countries only? US versus Europe?)
I am missing information on group size/ education of the instructors/ presence and process of debriefing.
Discussion:
The autors discuss the secondary goals of the study.
I am missing a clear statement of limitations.
Reviewer 2 Report
There are few language and grammatical errors like line 20-21 of abstract. It is mentioned that the 'technical skills were more frequently trained'. How the skills can be trained. There is similar error in line 60 and 76 of 'Introduction' and heading 3.2.3 of Results.
Abstract
1. Apart from the error cited above, the number of publications obtained are quoted as 1.617. This figure cannot be in fraction.
Introduction
1. Please elaborate the references cited in the text in line 67.
2. Language corrections are needed in line 60 and 76 as mentioned before.
Material and methods
1. There is language error in line 79.
2. The figure should be cited correctly in line 88.
Discussion
1. The first paragraph (line 193-195) is mere reiteration of aim and result and is superfluous.
2. Even, line 196 is unnecessary.
3. Language error in line 202, 210z
4. Please cite the references in the text in full and not as figure in parentheses (see line 208).
5. Line 208-209 are unclear. Reference 12 is not a review article and is cited incorrectly.
6. There is lot of redundancy in this section. At places the statement or the conclusion of the statement is ambiguous. The Discussion need revision to make it more impactful.
Conclusion
1. This section is too vague. Only relevant findings should be mentioned to make this section succinct.
